# *BRCA1/2* Molecular Assay for Ovarian Cancer Patients: A Survey through Italian Departments of Oncology and Molecular and Genomic Diagnostic Laboratories

**DOI:** 10.3390/diagnostics9040146

**Published:** 2019-10-09

**Authors:** Ettore Capoluongo, Nicla La Verde, Massimo Barberis, Maria Angela Bella, Fiamma Buttitta, Paola Carrera, Nicoletta Colombo, Laura Cortesi, Maurizio Genuardi, Massimo Gion, Valentina Guarneri, Domenica Lorusso, Antonio Marchetti, Paolo Marchetti, Nicola Normanno, Barbara Pasini, Matilde Pensabene, Sandro Pignata, Paolo Radice, Enrico Ricevuto, Anna Sapino, Pierosandro Tagliaferri, Pierfrancesco Tassone, Chiara Trevisiol, Mauro Truini, Liliana Varesco, Antonio Russo, Stefania Gori

**Affiliations:** 1Italian Society of Clinical Biochemistry and Clinical Molecular Biology—Laboratory Medicine (SIBioC), 20126 Milano, Italy; 2Dipartimento di Medicina Molecolare e Biotecnologie Mediche, Università Federico II—CEINGE, Biotecnologie Avanzate, 80100 Napoli, Italy; 3Italian Society of Medical Oncology (AIOM), 20133 Milano, Italy; nicla.laverde@asst-fbf-sacco.it (N.L.V.); ma.bella198@gmail.com (M.A.B.); nicoletta.colombo@ieo.it (N.C.); cortesilaura67@gmail.com (L.C.); massimo.gion@aulss3.veneto.it (M.G.); valentina.guarneri@unipd.it (V.G.); kettalorusso@libero.it (D.L.); paolo.marchetti@uniroma1.it (P.M.); nicnorm@yahoo.com (N.N.); pensabenematilde@gmail.com (M.P.); sandro.pignata@gmail.com (S.P.); enrico.ricevuto@univaq.it (E.R.); tagliaferri@unicz.it (P.T.); tassone@unicz.it (P.T.); Chiara.Trevisiol@aulss3.veneto.it (C.T.); antonio.russo@unipa.it (A.R.); 4Oncologia Medica, ASST Fatebenefratelli Sacco, 20157 Milano, Italy; 5Italian Society of Pathology-Italian Division of the International Academy of Pathology (SIAPEC-IAP), 20162 Milano, Italy; Massimo.Barberis@ieo.it (M.B.); fbuttitta@unich.it (F.B.); amarchetti@unich.it (A.M.); 6Histopathology and Molecular Diagnostics Unit, Istituto Europeo di Oncologia (IEO), 20122 Milano, Italy; mauroa.truini@gmail.com; 7Unità di Oncologia Medica Ospedale Maggiore di Parma, 43126 Parma, Italy; 8Dipartimento di Scienze Mediche, Orali e Biotecnologiche, Università di Chieti, 66100 Chieti, Italy; 9Italian Society of Human Genetics (SIGU), 20126 Milano, Italy; carrera.paola@hsr.it (P.C.); maurizio.genuardi@unicatt.it (M.G.); barbara.pasini@unito.it (B.P.); Paolo.Radice@istitutotumori.mi.it (P.R.); liliana.varesco@hsanmartino.it (L.V.); 10Laboratorio di Biologia Molecolare Clinica e Citogenetica, IRCCS Ospedale San Raffaele, 20132 Milano, Italy; 11Unità di Ginecologia Oncologica Medica, Istituto Europeo di Oncologia (IEO), 20122 Milano, Italy; 12Department of Oncology and Haematology, Genetic Oncology Unit, University Hospital of Modena, 41125 Modena, Italy; 13Fondazione Policlinico Universitario A. Gemelli IRCCS, UOC Genetica Medica, 00168 Rome, Italy; 14Istituto di Medicina Genomica, Università Cattolica del Sacro Cuore, 00167 Rome, Italy; 15Dipartimento di Scienze Chirurgiche, Oncologiche e Gastroenterologiche, Istituto Oncologico Veneto IRCCS, Oncologia Medica, 35128 Padova, Italy; 16Clinical Trial Center, Fondazione Policlinico Universitario Agostino Gemelli IRCCSC, 00168 Rome, Italy; 17UOC di Anatomia Patologica Asl 02 Lanciano-Vasto-Chieti: Università degli Studi “G. D’Annuncio”Chieti-Pescara, 66100 Chieti, Italy; 18Dipartimento di Scienze Oncologiche Policlinico Umberto I di Roma, 00161 Rome, Italuy; 19Cell Biology and Biotherapy Unit, Istiuto Nazionale Tumori “Fondazione G. Pascale”-IRCCS, 80131 Naples, Italy; 20Department of Medical Science, University of Turin, 10124 Turin, Italy; 21Department of Clinical Medicine and Surgery, University of Naples Federico II, 80131 Naples, Italy; 22Department of Urology and Gynecology, Istiuto Nazionale Tumori IRCCS “Fondazione G. Pascale”, 80131 Napoli, Italy; 23Unit of Molecular Bases of Genetic Risk and Genetic Testing, Department of Research, Fondazione IRCCS Istituto Nazionale dei Tumori, 20133 Milano, Italy; 24Oncology Network ASL1 Abruzzo, Oncology Territorial Care Unit, Division of Medical Oncology, Department of Biotechnological & Applied Clinical Sciences, University of L’Aquila, 67100 L’Aquila, Italy; 25Candiolo Cancer Institute-FPO-IRCCS, 10060 Candiolo (To), Italy; 26Department of Medical Sciences, University of Turin, 10126 Turin, Italy; 27Department of Experimental & Clinical Medicine, Magna Graecia University, Salvatore Venuta University Campus, 88100 Catanzaro, Italy; 28Translational Medical Oncology Unit, Department of Experimental and Clinical Medicine, Magna Græcia University and Cancer Center, Campus Salvatore Venuta, 88100 Catanzaro, Italy; 29Veneto Institute of Oncology IOV—IRCCS, 35128 Padua, Italy; 30Pathological Anatomy Histology & Cytogenetics, Niguarda Cancer Center, Niguarda Ca’ Granda Hospital, 20162 Milan, Italy; 31Unit of Hereditary Cancers, IRCCS AOU San Martino—IST, 16132 Genoa, Italy; 32Department of Surgical, Oncological & Oral Sciences, Section of Medical Oncology, University of Palermo, 90133 Palermo, Italy; 33Oncology Department, IRCCS Sacro Cuore Don Calabria, 37024 Negrar, Verona, Italy; stefania.gori@sacrocuore.it

**Keywords:** NGS, *BRCA1/2* assays, somatic BRCA, PARP-1i

## Abstract

In Italy, 5200 new ovarian cancers were diagnosed in 2018, highlighting an increasing need to test women for *BRCA1/2*. The number of labs offering this test is continuously increasing. The aim of this study was to show the results coming from the intersociety survey coordinated by four different Clinical and Laboratory Italian Scientific Societies (AIOM, SIAPEC-IAP, SIBIOC, and SIGU). A multidisciplinary team belonging to the four scientific societies drew up two different questionnaires: One was targeted toward all Italian Departments of Medical Oncology, and the second toward laboratories of clinical molecular biology. This survey was implemented from September 2017 to March 2018. Seventy-seven out of 305 (25%) Departments of Medical Oncology filled our survey form. Indeed, 59 molecular laboratories were invited. A total of 41 laboratories (70%) filled in the questionnaire. From 2014 to 2017, 16 new molecular laboratories were activated. A total of 12,559 tests were performed in the year 2016, with a mean of 339 tests and a median of 254 tests per laboratory, showing a glimpse of an extreme low number of tests performed per year by some laboratories. In terms of the type and number of professionals involved in the pre- and post-test counseling, results among the onco-genetic team were heterogeneous. Our data show that the number of laboratories providing *BRCA1/2* germline assays is significantly increased with further implementation of the somatic test coming soon. The harmonization of the complete laboratory diagnostic path should be encouraged, particularly in order to reduce the gap between laboratories with high and low throughput.

## 1. Introduction

About 10–15% of ovarian cancers are diagnosed in patients who have a hereditary breast and ovarian cancer (HBOC) syndrome because they are carriers of a *BRCA1/2* pathogenic variant (PV) [1]. In Italy, 5200 new ovarian cancers were diagnosed in 2018, with an overall survival of five years at 39%. The incidence trend appears to be decreasing in a statistically significant manner of −0.8% per year and does not present a north–south gradient in the country. The new cases, per 100,000 women/year, amount to 15.7%, 15.9%, and 13.8%, in the northern, central, and southern Italian regions, respectively [2]. Identifying *BRCA1/2* PV carriers is crucial to offer appropriate care and preventive programs for both cancer patients and their healthy relatives. The evaluation of *BRCA1/2* status can be performed on: (a) Blood samples, to identify a germline PV that is useful to diagnose a HBOC and to identify individuals at high risk; (b) Ovarian cancer specimens (fresh, frozen, or formalin-fixed paraffin embedded–FFPE) to recognize both somatic and germline PVs that allow patients to be eligible to specific treatments, such as Poly (ADP-ribose) polymerase1 (PARP1)-inhibitors [3]. The recognition of a PV on a tumor sample prompts reflex testing on blood in order to distinguish between germline and somatic variants. At this stage, the evaluation of large genomic rearrangements in *BRCA1/2* genes requires specific methods and can still be challenging, especially in tissue samples [3,4].

The fraction of individuals with a history of breast or ovarian cancer who meet *BRCA1/2* testing criteria, such as those devised by the National Cancer Comprehensive Network (https://www.nccn.org), was still limited, corresponding to less than 10–20%, between 1999–2013 [5]. Among the many explanations for such a low rate of testing, one was attributed to the lack of uniform geographic distribution of board-certified genetic counsellors [6].

The present study is the result of the collaboration of four scientific societies: The Italian Association of Medical Oncology (Associazione Italiana di Oncologia Medica, AIOM), the Italian Society of Human Genetics (Società Italiana di Genetica Umana, SIGU), the Società Italiana di Biochimica Clinica e Biologia Molecolare Clinica (SIBIOC), and the Italian Society of Pathology-Italian Division of the International Academy of Pathology (Società Italiana di Anatomia Patologica e di Citopatologia Diagnostica–Divisione Italiana della International Academy of Pathology, SIAPEC–IAP). All of them include professionals involved with *BRCA1/2* testing, either directly or indirectly (AIOM members are involved with test prescription and treatment/prevention). In 2015, they started to collaborate for the implementation of the test among Italian ovarian cancer patients and have produced national recommendations on this matter [7]. In 2017, the working group decided to perform a survey sending two different questionnaires: The first to the Departments of Medical Oncology, and the second to the Diagnostic Laboratories performing the *BRCA1/2* assay in order to gain information about the pathways that ovarian cancer patients follow in Italy to undergo these molecular tests.

## 2. Materials and Methods

The aim of this study was to survey the pathway that an ovarian cancer patient follows to undergo a *BRCA* test, germline and/or somatic test, including pre- and post-test genetic counseling.

Five components of the process on which the survey was focused were identified:
Core organization of the *BRCA1/2* testing process for ovarian cancer patients;Pipelines that were most frequently followed to request the test and which professionals are mainly involved;The testing approaches followed (somatic vs. germline testing) depending on the type of setting;The process of the referral to genetic counseling of both patients and healthy family members takes place;The volume of activity of the diagnostic laboratories and their procedures for the execution of the *BRCA1/2* assay.

A team composed of oncologists, gynecological oncologists, clinical geneticists, diagnostic laboratory experts (including molecular biologists/geneticists/pathologists), belonging to the four scientific societies (AIOM–SIGU–SIBIOC–SIAPEC-IAP) set up two different questionnaires: (1) A 13-item questionnaire to be submitted to all Italian Departments of Medical Oncology (Appendix A) recorded in the AIOM 2017 White Paper (8); (2) A 15 items questionnaire (Appendix A) to be submitted to diagnostic laboratories performing germline *BRCA1/2* tests.

The questionnaire for Oncologic Centers was published online on the AIOM website (http://www.aiom.it) in a reserved section and was only accessible through a direct link sent by email. After the first invitation, three further reminders were sent in order to increase the rate of response. The online survey was created using the GoogleDocs™ online surveys maker (https://docs.google.com).

The questionnaire for the laboratories was sent by email to the diagnostic laboratories. Like the previously described questionnaire, a link to the AIOM website was embedded in the email. Information about the type of technologies were collected about the use of next-generation sequencing (NGS), Multiplex Amplicon Quantification (MAQ), Multiplex Ligation-dependent Probe Amplification (MLPA), type of CNV analysis, Sanger sequencing, denaturing high-performance liquid chromatography (DHPLC), and quantitative multiplex PCR of short fluorescent fragments (QMPSF), in order to verify the capability of each laboratory.

Absolute frequencies and percentages were collected and organized with Microsoft Excel^TM^ in a separate manner for the two questionnaires. According to the exploratory intent of the survey, no formal statistical hypothesis was prespecified and no sample size was predefined.

## 3. Results

### 3.1. Survey on Departments of Oncology

From September 2017 to March 2018, 305 Departments of Medical Oncology registered in the AIOM 2017 White Paper [8] were invited to participate in a 13-item questionnaire. A total of 77 departments (25%) filled our survey form: 33 (50%) from northern Italy, 15 (20%) from central Italy, and 23 from southern Italy and the islands (30%). One center was located in the Republic of San Marino.

### 3.2. Genetic Units: Presence Inside the Hospitals and Their Organization

Regarding the first question about the presence in the hospital of a genetic unit/ambulatory where counseling could be performed, 74% of centers (57/77 centers) answered that they have one, while the remaining 26% (*n* = 20) send the patients to other hospitals for genetic counseling. About 79% of the centers located inside the hospital have an official recognition from the hospital board of management or from the regional authorities.

The executive responsibility of the counseling office is held by the oncologist in 47% of the centers and by the clinical geneticist in 39%.

Different professionals work inside the cancer genetics units. They are organized in different teams, as reported in Figure 1, where different professionals are involved in every specific hospital care units: Clinical geneticists counsellors (63%), biologists specialized in human genetics who were skilled in counselling (26%), oncologists (53%), surgeons (7%), nurses (12%), and psychologists (26%). As reported in Figure 1, only in 34% of the teams the counseling was performed by a medical geneticist or a biologist specialized in genetics as genetic counselor, while in 12%, it was done by the oncologist. A complete team, with a medical geneticist and a medical oncologist, was present in 14% of the cases.

### 3.3. Germline and Somatic BRCA1/2 Testing

In 39% of the 77 centers, ovarian cancer patients undergo germline *BRCA1/2* testing without previous genetic counselling and the test is prescribed directly by the oncologist. In 60% of the centers, patients are screened with the somatic test without prior genetic counselling. In these cases, the *BRCA1/2* test requests are carried out by the oncologist in 96% of the cases. Regarding the question: “In your centre, for a patient with ovarian cancer, do you request a germline or somatic *BRCA1/2* as a first step?”, 19% of centers answered that the first test prescription was for somatic testing, 69% for germline, and 12% did not answer. About 47% of the cases resulting negative at the first germline BRCA test were further assayed at the somatic level in order verify the patient’s eligibility to PARP-inhibitors. In most centers (87%), patients who resulted positive at the *BRCA1/2* somatic test were also re-tested at the germline level.

### 3.4. Genetic Counselling and Genetic Testing for Relatives of PV Carries

Once a diagnosis of *BRCA1/2* related HBOC was made for an ovarian cancer patient, family members are assessed to search for the family specific high-risk gene variant. Overall, 64% of the centers referred the relatives to a path of genetic counselling within the hospital itself. On the other hand, in 36% of the centers, relatives were sent to other academic or hospital centers to this purpose.

### 3.5. Survey on Diagnostic Laboratories

From September 2017 to March 2018, 59 Diagnostic Laboratories of Molecular Biology/Medical Genetics were invited to participate to complete a 15-item questionnaire. A total of 41 laboratories (70%) filled in the questionnaire: 21 (51%) from northern Italy, 7 (17%) from central Italy, and 13 from southern Italy and the islands (32%).

### 3.6. Duration of BRCA1/2 Testing Activities and Types of Germline Assays

Figure 2 shows the number of laboratories that were progressively able to provide BRCA assay since the germline *BRCA1/2* test was introduced. As shown in the figure, the centers were activated starting from 1995 onwards, with a significant increase from 2014. In fact, from 2014 to 2017, 16 new centers, representing 39% of the structures, were activated.

### 3.7. Technologies and Kit Employed for Germline BRCA1/2 Test

Different approaches, mainly the NGS-based technologies, were used for *BRCA1/2* screening (Table 1). Different types of kits were employed: 50% of the laboratories use CE-IVD commercial kits, 30% use RUO commercial kits adapted and internally validated for diagnostics, and 14% use a homemade pipeline, while 6% did not provide any details.

### 3.8. Yearly Volume of Germline BRCA1/2 Testing

In order to evaluate the activity of the laboratories, each laboratory was asked to reply to some questions about the total number of assays performed in the sample year 2016. As reported in Table 2, we collected all the data which were completely available at the time of the survey. Notably, a total of 12,559 tests were performed in 2016, with a mean of 339 and a median of 254 (range: 17–1660) per single laboratory. A total of 10,239 tests were performed to screen the whole *BRCA1/2* genes on cancer patients (all cancers) in 37 centers, with a mean of 277 and a median of 191 per laboratory, while a test for a known familiar variant was reported in 2738 patients, with a mean of 74 and median of 48.

### 3.9. Somatic BRCA1/2 Testing on Tumor Specimens

Overall, 28 out of 41 centers (68%) answered that they were carrying out somatic tests in addition to germline tests. Four started with setting up in 2015, 11 in 2016, and 13 in 2017 years, respectively. In all laboratories, a pathologist reviewed the tumor specimen before molecular *BRCA1/2* assay. The latter was performed on FFPE specimens in 96% of the cases, while only one laboratory reported the use of frozen tumor samples. Finally, both procedures (somatic and germline) were available in 25% of the centers. Regarding the running pipelines, 51% used a CE-IVD commercial kit, 39% used RUO commercial reagents, and 10% used a homemade pipeline, depending on the technological infrastructure present into each laboratory. In 2016, the 28 centers reported 699 somatic *BRCA1/2* tests performed on different tumors (Table 3), with a range between 5 and 200 samples per laboratory. Of these, 573 were performed on ovarian cancer specimens, with a range between 4 and 180.

### 3.10. Variant Classification and Interpretation

The largest part of the centers used two classification systems for reporting sequence variants: the IARC (73%) [9] and the AMG/AMP (54%) [10] scheme. Variants interpretation was carried out using either the ENIGMA (https://enigmaconsortium.org/wp-content/uploads/2018/10/ENIGMA_Rules_2017-06-29-v2.5.1.pdf) or the ACMG/AMP [10] guidelines (88% and 24% of centers, respectively).

## 4. Discussion

This survey was initiated as a result of a collaborative working group of four scientific societies with the aim of understanding the existing resources of Italian molecular and genomic diagnostic laboratories to perform and report *BRCA1/2* testing, both germline and somatic, in ovarian cancer patients. Overall, Italian oncology departments were asked to describe the care pathways followed by the patients to obtain a germline or somatic *BRCA1/2* test when not available in the same institution. On the other hand, molecular laboratories described their volumes of activity, techniques employed, and organizations. The adherence to the survey, although sufficient to provide a picture of the current situation was not completely satisfactory as to the response rate of the oncologists (only 25%), while it can considered adequate for the laboratories (70%).

Interestingly, the cancer centers and the laboratories which have responded to the surveys were mainly located in the north of the country, although the incidence of ovarian cancer was similar throughout the country. This probably reflects the heterogeneity of the distribution of cancer centers, the different organization of regional health systems, and the number of active and authorized laboratories at the time of survey. We further underline that the Italian regional systems are sometimes differently organized and depend on the population living in each region and the financial budget assigned to cover diagnostic tests. These conditions can also determine the differences in the access to BRCA test [2]. Finally, we cannot exclude the influence of financial imbalances of some Italian regions deriving from the economic crisis, as reported by de Belvis et al. [11].

On the other hand, it is of note that 79% of cancer centers have a genetic counseling service inside the hospital. The larger adherence to this survey was obtained from better organized centers routinely managing the entire paths related to HBOC, potentially leading to a selection bias. In support of this hypothesis, there is another important finding: In 39% of the centers a “mini-counseling” was adopted as a preliminary step. This is a shortened procedure, often administered by the gynecological oncologists who initiate the path of germline *BRCA1/2* testing, mainly focusing on therapeutic implications of the test and postponing a thorough discussion to second level genetic counseling, offered generally in presence of a positive BRCA result, which is performed by the geneticist. As in other countries, Italy has a shortage of medical geneticists. Therefore, the use of mini-counseling allows for the reduction of the bottleneck related to the long waiting lists for the genetic counseling [12]. These results remark the fact that Italy has fewer clinical geneticists or other health professionals trained in genetic counseling than what is needed. Therefore, as established in our Italian recommendations [7], the trained oncologists and gynecological oncologists can directly prescribe the test and perform a sort of preliminary counseling, reserving the extended counseling only to those patients who are carriers of a PV.

In this survey, another interesting observation is the heterogeneity of the members of the genetic counselling teams. In fact, in some centers, there are up to six professionals (oncologist, gynecologist, a specialist in medical genetics, psychologist, surgeon, nurse), while in others, only the oncologist or gynecological oncologist carries out mini-counseling. This could reflect the lack of specific national rules about the organization and composition of the team in the oncogenic boards. At this time, there are only some regional provisions that regulate or, better, suggest how this should work or should be organized [2,8,13,14].

Data from the present survey underline a significant rise in the number of centers providing the *BRCA1/2* assay, which highlights the progressive interest over the last 20 years in HBOC and its increasing clinical relevance. In fact, the high number of germline tests performed on both breast and ovarian cancers is related to the greater attention of medical specialists (oncologists and gynecologists) in selecting patients who could be carriers of PVs [1,3]. Further implementation of the somatic test is expected for the new therapeutic perspectives related to PARP inhibitors [15,16,17]. In fact, in 2016, 2648 germline tests and 573 somatic tests were performed in ovarian cancer patients. In the near future, an increase in *BRCA1/2* somatic test administration will probably occur, due to the need to administer PARP-inhibitors as first-line drugs [12,15,16,17]. Moreover, a better understanding of the mechanisms of resistance to PARP inhibitors is critical for the optimum strategy of PARP inhibitor. This may best be addressed by careful analysis of tumor samples from patients whose disease has progressed on PARP inhibitor therapy [17,18]. In this regard, our results show that some laboratories performed a very low number of tests per year. Importantly, the accuracy of the test can be dramatically influenced by the low volume of samples processed by many centers, particularly for the somatic assay.

The complexity of the *BRCA* assay, although not so difficult to manage in terms of on bench procedures, indeed requires a robust interpretation pipeline. The high quality of data (including copy number variation evaluation) can be ensured only managing an adequately large number of samples. In fact, laboratories processing many samples per week can gain a higher level of expertise compared to those running only few samples per year. Furthermore, the latter cannot fix the turnaround time if they have to wait a long time to collect a sufficient number of samples to completely fill the sequencer flow cell. This situation can be considered not only time-consuming, but also not proficient in terms of cost per test. Finally, classification systems for reporting sequence variants among the different centers were not harmonized because each lab used one or two different tools.

In conclusion, this first real-life survey can offer a picture of the current Italian situation for the *BRCA1/2* test in ovarian cancer. It can be a useful starting point to tightly collaborate with both institutions and patient associations in order to implement the diagnostic path and harmonize laboratory procedures across the national regions, including participation in external quality control schemes.

For the above-mentioned reasons, the scientific intersociety committee has recently published updated recommendations for the implementation of BRCA testing in ovarian cancer patients and their relatives [12]. We believe that our paper could be helpful for implementation of BRCA testing, particularly considering the increasing use for the assessment of PARP-1 sensitivity in other cancer entities, such as pancreatic and prostate cancers [15,16,17,18].

## Figures and Tables

**Figure 1 diagnostics-09-00146-f001:**
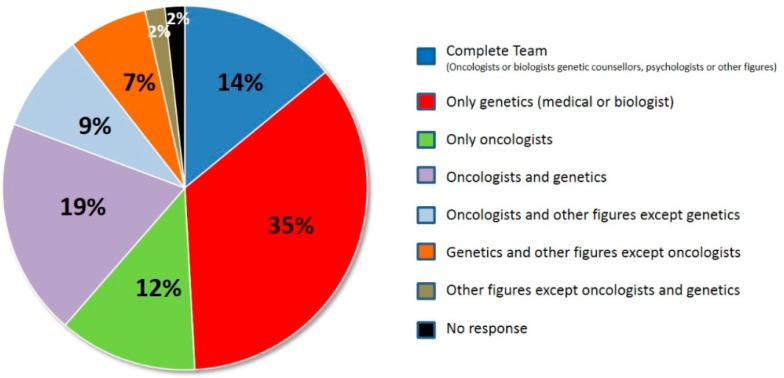
Professionals working within the team of the genetic counseling office.

**Figure 2 diagnostics-09-00146-f002:**
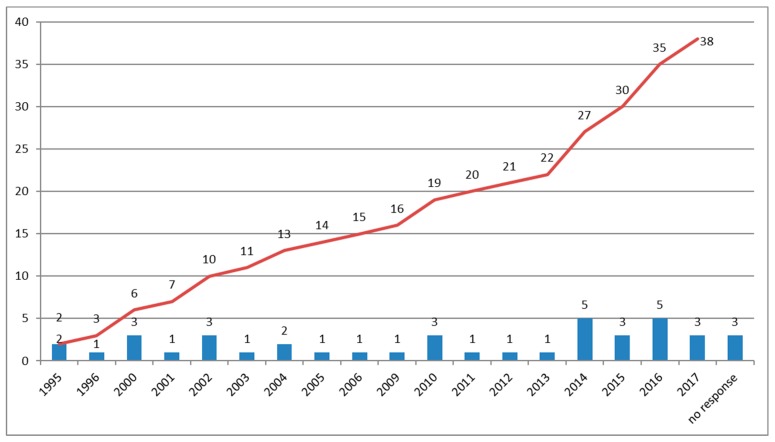
Number of new diagnostic laboratories activated per single year and cumulative number of new laboratories from 1995 to 2017. Red line: total number of labs per year; blue line: new labs starting with *BRCA*1/2 testing per year.

**Table 1 diagnostics-09-00146-t001:** Molecular techniques used to identify germline *BRCA1/2* pathogenic variant (PV).

N Lab	% Lab	Pipelines
3	7.3%	NGS
1	2.3%	NGS, MAQ
4	9.8%	NGS, MLPA
1	2.3%	NGS, ION TORRENT analysis of CNV
3	7.3%	NGS, Sanger
1	2.3%	NGS, Sanger, MAQ
21	51.2%	NGS, Sanger, MLPA
1	2.3%	NGS, Sanger, MLPA, NGS + MLPA * + DHPLC **
3	7.3%	Sanger, MLPA
1	2.3%	Sanger, QMPSF (for re-arrangements)
*2*	5.8%	no response

* to identify a Sanger unknown mutation; ** to identify a pathogenic variants in a family already known; NGS: Next Generation Sequencing. MAQ: Multiplex Amplicon Quantification; MPA: Multiplex Amplicon Quantification; QMPSF: Quantitative Multiplex PCR of Short Fluorescent Fragments.

**Table 2 diagnostics-09-00146-t002:** Germline *BRCA1/2* assay performed by molecular laboratories in 2016 in different cancer settings.

Results	Test Performed in 2016	Test Performed in Breast Cancer Patients	Test Performed in Ovarian Cancer Patients	Test Performed in Other Cancer (No Breast, No Ovarian) °	Test Performed in Healthy Patients
Number of test	12,559	6313	2648	172	1712
Mean (range)	339	175	76	7	54
Median	254 (17–1660)	136 (8–643)	35 (5–1000)	2 (1–28)	22 (4–500)
Centers responding to the questionnaires (out of 41)	37	36	35	26	32

° Laboratories provided other molecular tests covering Lynch syndrome, gastric, prostate, and pancreatic cancers, respectively.

**Table 3 diagnostics-09-00146-t003:** Somatic *BRCA1/2* test performed in 2016 in different cancer patients.

Results	Number of Tests Performed in 2016	Test in Ovarian Cancer Patients	Test in Breast Cancer Patients	Test in Other Cancers °
Total	699	573	76	50
Mean	30	32	6	5
Median (range)	19 (0–200)	20 (5–180)	0 (0–40)	0 (0–25)

° Laboratories also performed molecular tests on pancreatic and prostate cancers.

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
