# Peer review of "BRCA1/2 Molecular Assay for Ovarian Cancer Patients: A Survey through Italian Departments of Oncology and Molecular and Genomic Diagnostic Laboratories"

_diagnostics, 2019, doi:10.3390/diagnostics9040146_

Round 1

Reviewer 1 Report

Dear Editor,

Thank you for giving me the opportunity to review the manuscript submitted by Ettore Capoluongo et al entitled “BRCA1/2 molecular assay for Ovarian Cancer Patients: a survey through Italian departments of oncology and laboratories of clinical molecular biology” for consideration of publication in Diagnostics.

The manuscript describes the results of a study performed through application of a questionnaire to all involved centres in Italy asking the modality of BRCA testing mainly in ovarian cancer patients but also in other cancer entities. This descriptive study is interesting since BRCA testing is getting increasingly important not only in breast and ovarian cancer patients but also in pancreatic cancer patients (see results from the POLO trial) and perhaps in the future also in biliary tract cancer patients. Thus, getting an idea on the coverage of this molecular test in a country such as Italy is informative also for other countries. Hence, I think that this work merits publication in diagnostics provided that the authors make a thorough revision of many formal and linguistic errors throughout the manuscript. A lot of these errors are responsibly for a certain incomprehensibility of this manscript.

Minor points:

Abstract section: What do the authors mean with the sentence “Our data underline as the number of BRCA1/2 germline assays is significantly increased with further implementation of the somatic test coming soon” please rewrite the sentence with comprehensive wording. This is only one of many sentences throughout the manuscript that are not comprehensible. Table 2 The authors should specify what other cancer were tested (n=172) Table 3 The authors should specify what other cancer were tested Discussion section: the authors should include the information that BRCA testing could become important in other cancer entities and that the present data could be helpful for implementation of testing in these cancers too. Figure 2: please remove and correct the words “non risponde” In Material and Methods section the authors should explain the different methods used for BRCA testing presented in Table 1 (NGS, MAQ, MLPA, ION TORRENT analysy on CNV …) Line 102: there is probably a reference missing (see brackets)?? The authors should in any case make a revision of the manuscript by a native speaker

Author Response

Dear Editor,

Thank you for giving us the opportunity to revise and resubmit our paper (diagnostics-573360) entitled : BRCA1/2 molecular assay for Ovarian Cancer Patients: a survey through Italian departments of oncology and molecular and genomic diagnostic laboratories.

We have reviewed the manuscript following the Referees’ suggestions and comments.

Please, find below the response to reviewers. All changes are reported red in colour within the submitted version.

Thank you very much for your attention.

Regards

Prof. Ettore Capoluongo

REW1

Minor points:

Abstract section: What do the authors mean with the sentence “Our data underline as the number of BRCA1/2 germline assays is significantly increased with further implementation of the somatic test coming soon” please rewrite the sentence with comprehensive wording. This is only one of many sentences throughout the manuscript that are not comprehensible.

Response: We agree with need to clarify. we have slightly changed the sentence. 

Table 2 The authors should specify what other cancer were tested (n=172)

Response. We have detailed the type of cancer test in the footnotes to table 2.

Table 3 The authors should specify what other cancer were tested

Response. We have detailed the type of cancer test in the footnotes to table 3

 Discussion section: the authors should include the information that BRCA testing could become important in other cancer entities and that the present data could be helpful for implementation of testing in these cancers too.

Response: thanks for the suggestion. We have included a specific sentence guided by the reviewer’s  indication.

Figure 2: please remove and correct the words “non risponde”

Response: Figure 2 has been amended as requested.

In Material and Methods section the authors should explain the different methods used for BRCA testing presented in Table 1 (NGS, MAQ, MLPA, ION TORRENT analysy on CNV …)

Response: M&M section has been amended as requested

Line 102: there is probably a reference missing (see brackets)?? The authors should in any case make a revision of the manuscript by a native speaker

Response: we extensively reviewed the styple and language; we have also added the specific ref.

Reviewer 2 Report

This study survey the pathway that is followed in clinical practice for BRCA testing, germline and/or somatic, along with a pre and post-test genetic counselling. It is straightforward, well written, and concise. Definitely deserves to be published and is a valuable contribution to the “Diagnostics” Journal. Some minor flaws need to be addressed before publication.

Minor points:

[1] Lines 185-189: Do you believe that the discrepancy in the distribution of the cancer centres and laboratories between the north and south country could be correlated to the relevant differences in terms of the financial status, unemployment rate and other similar socioeconomics parameters?

Relevant reference: de Belvis AG, et al. The financial crisis in Italy: implications for the healthcare sector. Health Policy. 2012 Jun;106(1):10-6.

[2] Lines 208-210: Here, a reference should be added.

Relevant reference: Boussios S, et al. Combined Strategies with Poly (ADP-Ribose) Polymerase (PARP) Inhibitors for the Treatment of Ovarian Cancer: A Literature Review. Diagnostics (Basel). 2019 Aug 1;9(3). pii: E87.

[3] Sentence in lines 213-214: At that point, you should make a comment about PARP inhibitors’ resistance. Better understanding of the mechanisms of resistance to PARP inhibitors is critical for the optimum strategy of PARP inhibitor. This may best be addressed by careful analysis of tumour samples from patients whose disease has progressed  on  PARP inhibitor therapy.

Relevant reference: Boussios S, et al. PARP Inhibitors in Ovarian Cancer: The Route to "Ithaca". Diagnostics (Basel). 2019 May 18;9(2). pii: E55.

Author Response

Dear Editor,

Thank you for giving us the opportunity to revise and resubmit our paper (diagnostics-573360) entitled : BRCA1/2 molecular assay for Ovarian Cancer Patients: a survey through Italian departments of oncology and molecular and genomic diagnostic laboratories.

We have reviewed the manuscript following the Referees’ suggestions and comments.

Please, find below the response to reviewers. All changes are reported red in colour within the submitted version.

Thank you very much for your attention.

Regards

Prof. Ettore Capoluongo

REW2

This study survey the pathway that is followed in clinical practice for BRCA testing, germline and/or somatic, along with a pre and post-test genetic counselling. It is straightforward, well written, and concise. Definitely deserves to be published and is a valuable contribution to the “Diagnostics” Journal. Some minor flaws need to be addressed before publication.

 Minor points:

[1] Lines 185-189: Do you believe that the discrepancy in the distribution of the cancer centres and laboratories between the north and south country could be correlated to the relevant differences in terms of the financial status, unemployment rate and other similar socioeconomics parameters?

Response: thanks for this suggestion. We have reviewed and better clarified this sentence, also providing our view in within the discussion.

Relevant reference: de Belvis AG, et al. The financial crisis in Italy: implications for the healthcare sector. Health Policy. 2012 Jun;106(1):10-6.

Response: we have added sentence and reference within the text

[2] Lines 208-210: Here, a reference should be added.

Relevant reference: Boussios S, et al. Combined Strategies with Poly (ADP-Ribose) Polymerase (PARP) Inhibitors for the Treatment of Ovarian Cancer: A Literature Review. Diagnostics (Basel). 2019 Aug 1;9(3). pii: E87

Response. We have added the reference within the text (ref16)

[3] Sentence in lines 213-214: At that point, you should make a comment about PARP inhibitors’ resistance. Better understanding of the mechanisms of resistance to PARP inhibitors is critical for the optimum strategy of PARP inhibitor. This may best be addressed by careful analysis of tumour samples from patients whose disease has progressed  on  PARP inhibitor therapy

.Relevant reference: Boussios S, et al. PARP Inhibitors in Ovarian Cancer: The Route to "Ithaca". Diagnostics (Basel). 2019 May 18;9(2). pii: E55.

Response. We have added sentence and the relative  reference within the text (ref17)